# Imaging Neurodegenerative Metabolism in Amyotrophic Lateral Sclerosis with Hyperpolarized [1-13C]pyruvate MRI

**Nikolaj Bøgh** [1,*], **Christoffer Laustsen** [1], **Esben S. S. Hansen** [1], **Hatice Tankisi** [2], **Lotte B. Bertelsen** [1] **and Jakob U. Blicher** [3,4]

1   The MR Research Center, Department of Clinical Medicine, Aarhus University, 8200 Aarhus, Denmark; cl@clin.au.dk (C.L.); esben@clin.au.dk (E.S.S.H.); lotte@clin.au.dk (L.B.B.)
2   Department of Clinical Neurophysiology, Aarhus University Hospital, 8200 Aarhus, Denmark; hatitank@rm.dk
3   Center of Functionally Integrative Neuroscience, Aarhus University, 8200 Aarhus, Denmark; jbli@cfin.au.dk
4   Department of Neurology, Aalborg University Hospital, 9000 Aalborg, Denmark
*   Correspondence: nikolaj.boegh@clin.au.dk

**Abstract:** The cause of amyotrophic lateral sclerosis (ALS) is still unknown, and consequently, early diagnosis of the disease can be difficult and effective treatment is lacking. The pathology of ALS seems to involve specific disturbances in carbohydrate metabolism, which may be diagnostic and therapeutic targets. Magnetic resonance imaging (MRI) with hyperpolarized [1-13C]pyruvate is emerging as a technology for the evaluation of pathway-specific changes in the brain's metabolism. By imaging pyruvate and the lactate and bicarbonate it is metabolized into, the technology is sensitive to the metabolic changes of inflammation and mitochondrial dysfunction. In this study, we performed hyperpolarized MRI of a patient with newly diagnosed ALS. We found a lateralized difference in [1-13C]pyruvate-to-[1-13C]lactate exchange with no changes in exchange from [1-13C]pyruvate to 13C-bicarbonate. The 40% increase in [1-13C]pyruvate-to-[1-13C]lactate exchange corresponded with the patient's symptoms and presentation with upper-motor neuron affection and cortical hyperexcitability. The data presented here demonstrate the feasibility of performing hyperpolarized MRI in ALS. They indicate potential in pathway-specific imaging of dysfunctional carbohydrate metabolism in ALS, an enigmatic neurodegenerative disease.

**Keywords:** metabolic; magnetic resonance imaging; hyperpolarized; pyruvate; amyotrophic lateral sclerosis; neurodegeneration



## 1. Introduction

Magnetic resonance imaging (MRI) with hyperpolarized metabolically active molecules is an emerging technology that enables the imaging of specific metabolic pathways [1]. Hyperpolarization with dynamic nuclear polarization increases the signal of 13C-enriched molecules by four to five orders of magnitude. In this process, the high polarization of free electrons in a strong magnetic field at ~0.7 K is transferred to the 13C-spins in the molecule of interest using microwave irradiation [2]. Once hyperpolarized, the 13C-enriched molecules are detectable using the MR system. A wealth of molecules is being investigated as probes for various purposes. The probe closest to clinical application is hyperpolarized [1-13C]pyruvate. When hyperpolarized [1-13C]pyruvate is administered intravenously, its delivery to the brain and subsequent metabolism to lactate and bicarbonate can be imaged due to the chemical shift effect. This metabolism is conveyed by uptake through the monocarboxylate transporters over the blood-brain barrier and cell membranes as well as by intracellular metabolism catalyzed by the lactate dehydrogenase and the pyruvate dehydrogenase (Figure 1). The lactate and bicarbonate signals thus represent imaging readouts that are sensitive to blood–brain barrier uptake and subsequent glycolytic and oxidative metabolism. The technology is currently under clinical translation

with a particulate focus on cancer imaging [3]. Here, we present an initial experience on its feasibility and potential in neurodegenerative disease.

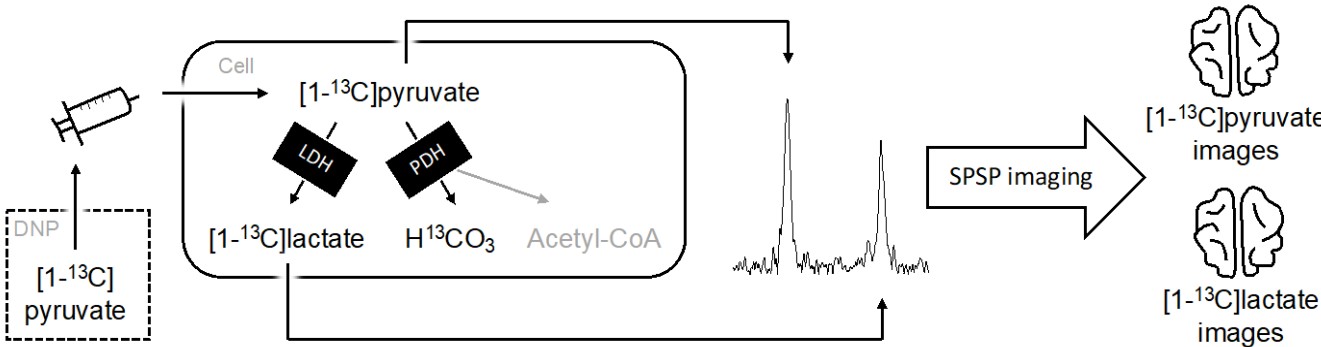

**Figure 1.** In hyperpolarized MRI, [1-$^{13}$C]pyruvate is brought to a state of polarization >10.000 times above the thermal equilibrium using dynamic nuclear polarization (DNP). Following hyperpolarization, the product is administered intravenously. In the target organ, the [1-$^{13}$C]pyruvate is taken up and metabolized to lactate or bicarbonate through the lactate dehydrogenase (LDH) or the pyruvate dehydrogenase (PDH), respectively. This yields three separate peaks that are shifted enough to be imaged separately using spectral-spatial (SPSP) imaging. The spectrum shown is from the patient of this case report. It was obtained after imaging where the signal-to-noise was too low to observe bicarbonate.

Albeit immense research efforts, amyotrophic lateral sclerosis (ALS) remains a poorly understood disease and a challenge for researchers and clinicians alike. The diagnostic process is long, requires many tests, and is encumbered with uncertainty as mimics must be considered [4]. The prognosis is dismal, with considerable morbidity and a few years of expected survival after diagnosis [5]. ALS is a neurodegenerative disease associated with the accumulation of TAR DNA-binding protein 43 (TDP43) in motor neurons [5]. However, the causes of this TDP43 proteinopathy are mostly unknown. The disease is considered multifactorial with several known environmental and genetic risk factors. Interestingly, several of these factors are involved in metabolism. For example, mutations in the genes coding for the superoxide dismutase 1 (SOD1) and the TDP43 are associated with mitochondrial dysfunction and oxidative stress [5]. Markers of energy deprivation are increased in motor neurons of ALS patients and correlate with proteinopathy [6]. Specifically, the intricate pathophysiology of ALS involves dysfunction of carbohydrate metabolism from uptake to utilization [7–9]. The metabolic dysfunction is not necessarily confined to neurons but likely also involves glial cells, which are thought to support neurons metabolically and which may be an overlooked element of ALS pathophysiology [7,10]. Importantly, these metabolic disturbances are emerging as therapeutic targets [11].

Clinically, fluorodeoxyglucose positron emission tomography ([$^{18}$F]FDG PET) can detect hypometabolic signatures that are useful in the diagnostic process, especially when dementia is considered. However, [$^{18}$F]FDG PET is unable to detect changes in pathways downstream of glucose uptake, and the observed hypometabolism may be caused by neuronal death rather than underlying metabolic dysfunction. As such, assessment of oxidative versus glycolytic metabolism with hyperpolarized [1-$^{13}$C]pyruvate MRI could advance our understanding of the roles of metabolism of ALS as well as serve as a biomarker in clinical studies targeting metabolic dysregulation.

## 2. Case

### 2.1. Clinical Presentation

A 59-year-old woman presented with bulbar onset ALS. Her initial symptoms were dyspnea, dysarthria, dysphagia, and a ten-kilogram weight loss in the last 12 months prior to the diagnosis. She had no other health issue except for a right shoulder biceps-tendinitis and essential thrombocytosis. She had no family history of ALS or dementia.

At the time of referral, clinical examination of the cranial nerves revealed tongue atrophy and fasciculations as well as a hyperactive jaw reflex. In the upper extremities, the force was normal, but the triceps, biceps, and brachioradialis reflexes were hyperactive, and there was a positive Hoffmann's reflex bilaterally. The blood samples were normal. The cerebrospinal fluid samples were normal except for elevated levels of Neurofilament light chain (4805 ng/L). Structural MRI of the brain was normal. Quantitative electromyography (EMG) showed signs of both acute (fibrillations and fasciculations) and chronic (increased Motor Unit Potential duration and amplitude) denervation in the tongue and the right biceps, left vastus medialis, and right tibialis anterior muscles. Transcranial magnetic stimulation (TMS) showed marked cortical disinhibition in the left motor cortex using both standard paired-pulse TMS as well as threshold tracking TMS [12]. Consequently, the patient was diagnosed with ALS. Riluzole treatment was initiated but was later discontinued due to gastrointestinal side effects.

At the time of scanning with hyperpolarized [1-$^{13}$C]pyruvate, the patient had developed a right-sided drop-foot and weakness in both hands. She had a Penn Upper Motor Neuron score of 9/32 (lower is better; [13]), a revised ALS Functional Rating Scale (ALSFRS-R) score of 37/48 (higher is better; [14]), and no signs of cognitive or behavioral deficits on the Edinburgh Cognitive and Behavioral ALS screen [15]. Collectively, her symptoms were dominated by right-sided and bulbar motor issues with considerable upper motor neuron pathology. She did not receive Riluzole at the time of MRI with hyperpolarized [1-$^{13}$C]pyruvate.

## 2.2. Hyperpolarized Pyruvate MRI

After informed consent, the patient underwent MRI with hyperpolarized [1-$^{13}$C]pyruvate following a protocol approved by the Danish Medicines Agency and the Committee on Health Research Ethics for Central Denmark (EudraCT 2020-000352-36). Imaging was performed on a 3T scanner (MR750, GE Healthcare, Chicago, IL, USA) using a $^{13}$C/$^1$H-tuned birdcage transceiver coil (PulseTeq, Surrey, UK). The patient was fully awake and not sedated for the examination. A basic proton exam was performed (Figure 2). This included a T1-weighted anatomical reference (2D fast spoiled gradient echo, echo time/repetition time = 2.3/163 ms, flip angle = 85°, matrix size = 256 × 256, field of view = 24 × 24 cm$^2$, slice thickness = 4 mm), perfusion imaging (3D pseudo-continuous arterial spin labeling with a spiral readout, in-plane resolution = 3.6 mm$^2$, slice thickness = 6 mm, post label delay = 2025 ms, scan time = 4 min 15 s), and spectroscopy (single-voxel point resolved spectroscopy, 2 × 2 × 2 cm$^3$ voxel placed at the hand knob, 8 averages, echo time/repetition time = 135/2000 ms). Shimming was performed as second-order shimming. There were no structural abnormalities. No apparent change in perfusion was observed (81.8 versus 82.6 mL/100 mL/min). The N-acetylaspartate/creatine ratio was 1.76 versus 1.59 in the left and right motor cortices, respectively.

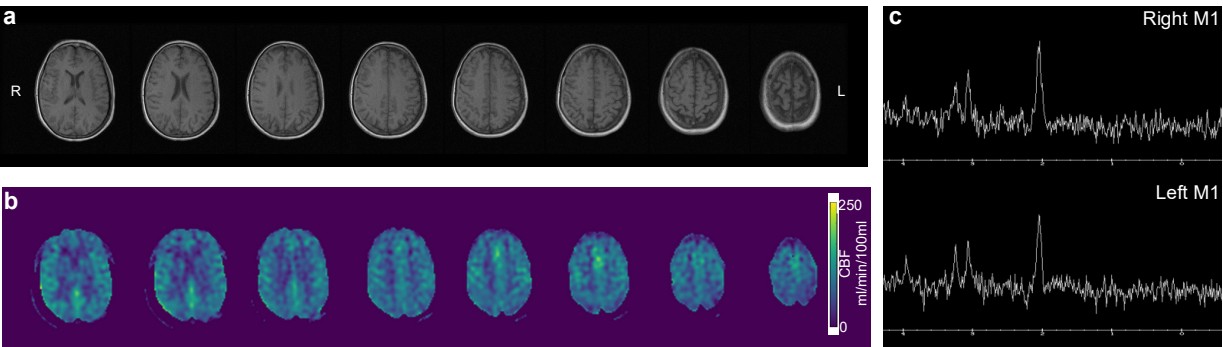

**Figure 2.** Routine magnetic resonance imaging with T1 weighted (**a**), arterial spin labeling perfusion imaging (**b**), and spectroscopy (**c**) revealed no apparent pathology.

For hyperpolarized MRI, the [1-$^{13}$C]pyruvic acid was polarized using commercial equipment (SPINlab, GE Healthcare) as previously described [16]. In short, 1.47 g of good manufacturing practice grade [1-$^{13}$C]pyruvic acid (Sigma-Aldrich, Søborg, Denmark) underwent dynamic nuclear polarization at 5 T and 0.7 K [2] with 15 mM AH111501 (Syncom, Groningen, The Netherlands). After hyperpolarization, the product was dissolved, buffered, filtered, and underwent quality control. The final product was administered to the patient at a dose of 0.43 mL/kg at 5 mL/s chased by 20 mL of saline at the same rate. Following IV administration of hyperpolarized [1-$^{13}$C]pyruvate, the [1-$^{13}$C]pyruvate and downstream [1-$^{13}$C]lactate and [$^{13}$C]bicarbonate resonances were imaged separately using spectral-spatial metabolite-selective excitation (flip angles = 12°/70°/70° for pyruvate, lactate, and bicarbonate, respectively; [17]). A dual-resolution spiral readout was employed (pyruvate resolution = 0.875 × 0.875 mm$^2$, metabolite resolution = 1.75 × 1.75 mm$^2$). In total, six slices of 2 cm thickness were acquired from the vertex and down. This yielded dynamic images of pyruvate, lactate, and bicarbonate with a time resolution of 2 s. The imaging was initiated immediately after the saline flush, and the total scan time for the $^{13}$C sequence was 80 s. The conversion of pyruvate to lactate ($k_{PL}$) and pyruvate to bicarbonate ($k_{PB}$) were quantified with a one-way metabolic exchange kinetic model (Figure 3). The employed model did not consider the pyruvate input curve, the relaxation rates were fixed, and 0.02 s$^{-1}$ and 0.005 s$^{-1}$ were used as starting values for $k_{PL}$ and $k_{PB}$, respectively. The modeling was performed with the Hyperpolarized MRI Toolbox (doi: 10.5281/zenodo.1198915; [18]). In addition to kinetic modeling, we computed the ratio of the lactate signal to the bicarbonate signal. Calibration of transmit power was performed on a glycerol head phantom with a 90° hard pulse. The center frequency was extrapolated from the proton frequency [19] and confirmed after imaging with a pulse-acquire spectrum of a single axial slice over the brain (soft pulse, 100 mm thickness, 30° degree flip angle, 2048 points over 5000 Hz).

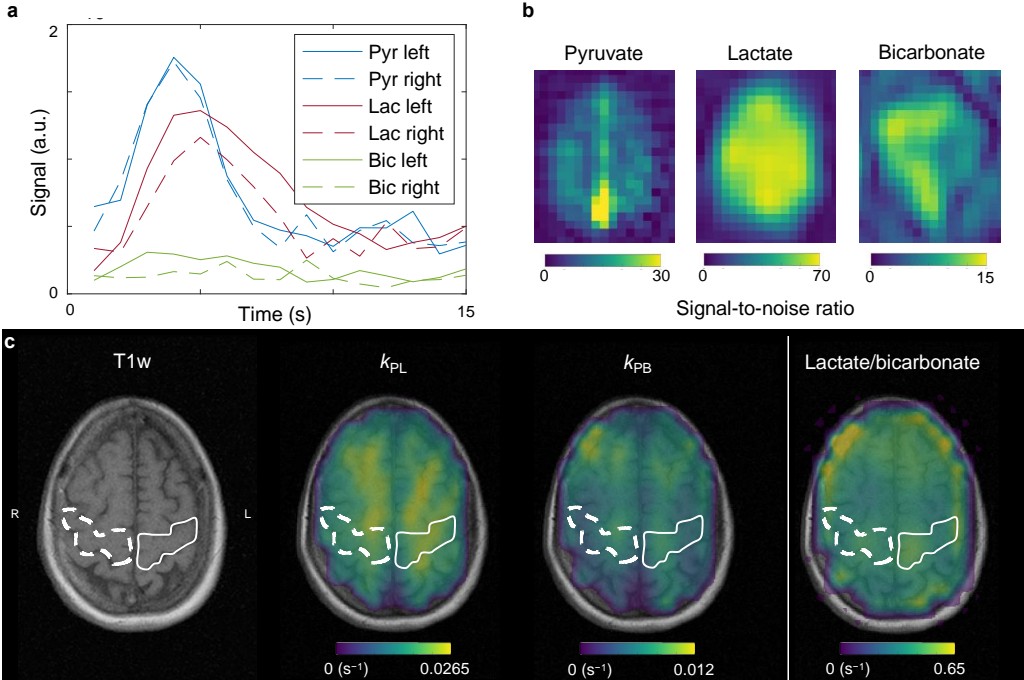

**Figure 3.** After administration of hyperpolarized [1-$^{13}$C]pyruvate, dynamic images were acquired, yielding time curves in the motor cortex as presented in (**a**). The signal-to-noise ratio summed over time is presented in (**b**). Kinetic fitting (**c**) of the dynamic data allows estimation of conversion of pyruvate to lactate ($k_{PL}$) and bicarbonate ($k_{PB}$). Further, the model-free ratio of the lactate to bicarbonate signals is shown. Increased conversion to lactate was observed in the left hand motor area compared to the right (solid versus dashed lines).

Using hyperpolarized [1-$^{13}$C]pyruvate MRI, we found that the left motor cortex displayed ~40 % larger conversion from pyruvate to lactate than the right ($k_{PL}$ = 0.023 s$^{-1}$ versus 0.014 s$^{-1}$). There was no change in $k_{PB}$ (0.0064 s$^{-1}$ versus 0.0065 s$^{-1}$). The lactate-to-bicarbonate ratios were 0.45 in the left hemisphere versus 0.4 in the right. The mean transit time of pyruvate, estimated as previously described [16], was 15.5 s in the left hemisphere and 19.6 s in the right hemisphere.

## 3. Discussion

To our knowledge, this is the first report of in vivo imaging of neurodegenerative cortical metabolism with hyperpolarized [1-$^{13}$C]pyruvate MRI in humans. The case seemingly displayed increased pyruvate-to-lactate conversion in the left motor area, corresponding with an increased symptom burden in the right hand and leg. Previous studies in healthy volunteers have found no lateralized differences in $k_{PL}$ [20]. We found no apparent changes in pyruvate-to-bicarbonate metabolism or perfusion from arterial spin labeling.

Preclinical research in a mouse model of multiple sclerosis suggests that conversion to lactate may be a marker of inflammatory metabolism [21]. As for many other neurodegenerative diseases, chronic neuroinflammation seems to be a pathological feature of ALS [22]. In addition to inflammatory cells, lactate is produced by astrocytes and oligodendroglia. This lactate is then excreted to be utilized by neurons for oxidation, according to the astrocyte-neuron lactate shuttle hypothesis [23–25]. A build-up of lactate could suggest the failure of this supportive mechanism. A third explanation might be the cortical hyperexcitability observed in ALS generally and this patient specifically [12], potentially leading to increased flux through the astrocyte-neuron lactate shuttle. The molecular and pathophysiological correlates of changes in pyruvate-to-lactate conversion could be explored in future basic and translational studies. Similar to the $k_{PL}$, the mean transit time of pyruvate was shorter in the left hemisphere than in the right hemisphere, while the perfusion measured with arterial spin labeling was similar between the two sides. One possible explanation for this may be increased uptake of pyruvate into reactive glial cells (i.e., astrocytes and microglia) followed by swift metabolism. This would cause a shorter apparent transit time. As glia cells may contribute significantly to ALS pathogenesis, further exploring this idea may prove fruitful [10]. A recent study from our group shows that hyperpolarized [1-$^{13}$C]pyruvate MRI might be able to provide unique insight into the metabolic interplay of neurons and glial cells [26].

Interestingly, the $k_{PB}$, a marker of mitochondrial metabolism, was unaltered between hemispheres in this case. This could suggest that hyperpolarized MRI is insensitive to subtle changes in mitochondrial function. One reason for this could be the low signal-to-noise ratio of bicarbonate in a hyperpolarization experiment. Several means exist to improve this, including optimized coil setups and flip angle schemes [26,27], and future work should explore if an improved signal-to-noise ratio will reveal any changes in the bicarbonate signal in neurodegenerative disease. An alternative explanation of similar $k_{PB}$ between hemispheres is that mitochondrial dysfunction might not have been present in this case. This could be due to the patient being in a relatively early stage of disease or due to a less prevalent role of mitochondrial dysfunction in non-SOD1 ALS than in the less prevalent but much-studied SOD1 model [5].

Naturally, the conclusions that can be drawn from a single case are limited. As such, we are unable to more than speculate about the slightly lower N-acetylaspartate/creatine ratio of the right M1 and its relation to the hyperpolarized data or the changes that could be expected in the more frontal parts of the brain in some ALS patients. Likewise, this patient presented with bulbar onset, and it is hard to evaluate the laterality of bulbar upper motor neuron symptoms. However, the lateralized differences that we observed in the hand and arm areas are of interest. Patients with ALS often present with lateralized symptoms, of which it can be difficult to assess the contributions of pathology in upper and lower motor neurons. Determining metabolic dysfunctions that may correlate with lateralized upper motor neuron pathology would thus be of clinical and research interest. Nevertheless,

more cases and comparison to healthy, age-matched controls is warranted to draw any clear conclusions. The data presented here merely suggest an avenue of exploration into enigmatic neurodegenerative diseases.

MRI with hyperpolarized [1-$^{13}$C]pyruvate is a technology under development. Since the initial clinical trial, an abundance of smaller trials has been published [3]. Now, larger single-center studies are ongoing, multisite trials are possible [28], and the technology is able to provide good quality data routinely. Throughput and availability are expected to increase as the process of making the hyperpolarized probes are further developed [29–31]. Looking beyond hyperpolarized [1-$^{13}$C]pyruvate, numerous other probes exist for investigations of other facets of metabolism [32]. This includes hyperpolarized 2-keto[1-$^{13}$C]isocaproate, which might be indicative of glutamate metabolism, and hyperpolarized [1-$^{13}$C]acetate, which is also metabolized by the brain, and several markers of oxidative stress. These are all, however, still at the preclinical stage, and [1-$^{13}$C]pyruvate is likely to be the probe that the field focuses the most on for the immediate future.

In conclusion, these results demonstrate the emerging potential of hyperpolarized MRI in the imaging of cortical pathology in ALS. Further investigations will evaluate the findings presented here as well as provide guidance on the potential usability of hyperpolarized MRI as a clinical and research tool.

**Author Contributions:** Conceptualization, N.B., C.L. and J.U.B.; methodology, N.B., C.L. and E.S.S.H.; investigation, N.B., E.S.S.H., H.T., L.B.B. and J.U.B.; writing—original draft preparation, N.B.; writing—review and editing, N.B., C.L., E.S.S.H., H.T., L.B.B. and J.U.B.; visualization, N.B.; supervision, C.L., E.S.S.H. and J.U.B.; funding acquisition, C.L. All authors have read and agreed to the published version of the manuscript.

**Funding:** This work was funded by the Lundbeck Foundation (grant number: R272-2017-4023).

**Institutional Review Board Statement:** The study was conducted in accordance with the Declaration of Helsinki, and approved by the Danish Medicines Agency and the Committee on Health Research Ethics for Central Denmark (EudraCT 2020-000352-36).

**Informed Consent Statement:** Informed consent was obtained from all subjects involved in the study.

**Data Availability Statement:** The data presented in this study are available on request from the corresponding author. The data are not publicly available due to GDPR regulations.

**Acknowledgments:** The authors would like to thank Tau Vendelboe, Duy Anh Dang, and Mette Dalgaard for their technical and laboratory support.

**Conflicts of Interest:** The authors declare no conflict of interest.

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
