# Peer review of "Imaging Neurodegenerative Metabolism in Amyotrophic Lateral Sclerosis with Hyperpolarized [1-13C]pyruvate MRI"

_tomography, doi:10.3390/tomography8030129_

Round 1

Reviewer 1 Report

This case report illustrates a possible feasibility of a new imaging method, hyperpolarized MRI with specific metabolism, for brain in amyotrophic lateral sclerosis (ALS). Antioxidative metabolic alteration, which have been hypothesized, was assessed using hyperpolarized brain MRI in qualitative mapping noninvasively.

The finding is encouraging for providing an opportunity for assessment of brain pathophysiology in ALS. However, there are significant concerns on some critical points. Below I address some points that should be addressed.

Major concerns

While the authors prefer that this methodological framework can be utilized in clinical settings, the advantage of this method should be validated in a more clinically oriented manner.

  1. The author should describe the rationality of applying this method to ALS more clearly and concisely based on previous evidence from a neurochemical and neuropathological standpoint.
  2. I believe that the interested readers would like to try to replicate the findings. Therefore, it would be a good idea to describe the detail of imaging parameters such as total imaging time, sedative procedures, or other preparation.
  3. The authors need to show time courses of the signal from regions of interests in the affected and the non-affected areas, because the k maps are too smoothed to identify the hemispheric differences.

Minor points

  1. The figures of results should have more appropriate color lookup table so that the interested readers can identify and understand the findings.
  2. Precise and appropriate placing of less black arrows would be preferable for better understanding of the present findings.

In summary, the authors should reconsider these points to make the report relevant and show how it can be clinically useful.

Reviewer 2 Report

This reviewer is very impressed that pyruvate conversion to lactate can be measured in a patient with ALS!  However, here are some suggestions to help improve the manuscript.

1) The authors need to provide a lot more details/figures on how C-13 is measured and coupled to protons.

2) the authors need to provide proof that the MRS signal is actually measuring C-13 and not just proton MRS.  Show us a figure proving the C-13 coupling to protons of both pyruvate and lactate, in other words, show us a spectrum of pyruvate and lactate edited spectrum.

Round 2

Reviewer 1 Report

The authors have addressed all my previous concerns with their new revision and new figures.  The manuscript is now ready for publication.

Reviewer 2 Report

The authors have addressed all my previous concerns with their new revision and new figures.  The manuscript is now ready for publication.